# Anti-Inflammatory, Antimicrobial, Antioxidant and Photoprotective Investigation of Red Propolis Extract as Sunscreen Formulation in Polawax Cream

**DOI:** 10.3390/ijms24065112

**Published:** 2023-03-07

**Authors:** Thalita Marcolan Valverde, Bruna Nayane Goncalves de Souza Soares, Andréa Mendes do Nascimento, Ângela Leão Andrade, Lucas Resende Dutra Sousa, Paula Melo de Abreu Vieira, Vagner Rodrigues Santos, Janaína Brandão Seibert, Tatiane Cristine Silva de Almeida, Caio Fabrini Rodrigues, Samantha Roberta Machado de Oliveira, Flaviano dos Santos Martins, Jeronimo Geraldo Ferreira Júnior, Viviane Martins Rebello dos Santos

**Affiliations:** 1Department of Morphology, Institute of Biological Sciences, Federal University of Minas Gerais (UFMG), Belo Horizonte 31270-901, MG, Brazil; 2Department of Chemistry, Federal University of Ouro Preto (UFOP), Ouro Preto 35400-000, MG, Brazil; 3Laboratory of Morphopathology, Center for Research in Biological Sciences, Federal University of Ouro Preto (UFOP), Ouro Preto 35400-000, MG, Brazil; 4Laboratory of Microbiology and Biomaterials, School of Dentistry, Federal University of Minas Gerais (UFMG), Belo Horizonte 31270-901, MG, Brazil; 5Natural Products Laboratory, Department of Chemistry, Federal University of São Carlos (UFSCAR), São Carlos 13565-905, SP, Brazil; 6Nuclear Technology Development Center, Belo Horizonte 6627, MG, Brazil; 7Department of Microbiology, Institute of Biological Sciences, Federal University of Minas Gerais (UFMG), Belo Horizonte 31270-901, MG, Brazil

**Keywords:** red propolis, photoprotective, antioxidant, ethanolic extracts, anti-inflammatory, chemical composition

## Abstract

Many activities have been described for propolis, including, antiviral, antibacterial, antifungal, anti-inflammatory, immunoregulatory, antioxidant and wound healing properties. Recently, propolis has been highlighted due to its potential application in the pharmaceutical and cosmetic industries, motivating a better understanding of its antioxidant and anti-inflammatory activities. Propolis and its main polyphenolic compounds presented high antioxidant activity, and effectiveness as broad spectrum UVB and UVA photoprotection sunscreens. Through a qualitative phytochemical screening, the ethanolic red propolis extracts (EEPV) (70% at room temperature and 70% at a hot temperature) presented a positive result for flavonoids and terpenoids. It presented an antioxidant activity for reducing 50% of DPPH of 17 and 12 μg/mL for extraction at room temperature and at a hot temperature, respectively. The UPLC-QTOF-MS/MS analysis allowed the annotation of 40 substances for EEPV-Heated and 42 substances for EEPV-Room Temperature. The IC_50_ results of the ABTS scavenging activity was 4.7 μg/mL for both extractions, at room temperature and at a hot temperature. Additionally, we also evaluated the cytotoxic profile of propolis extracts against macrophage (RAW 264.7 cells) and keratinocytes (HaCaT cells), which showed non-cytotoxic doses in cell viability assays even after a long period of exposure. In addition, propolis extracts showed antibacterial activity for Gram-positive bacteria (*Staphylococcus aureus* and *Staphylococcus epidermidis*), demonstrating potential biological activity for the creation of formulations aimed at disease control and prevention.

## 1. Introduction

Propolis has been used as a herbal medicine and several useful activities have been described for propolis, including antiviral, antibacterial, antifungal, anti-inflammatory, antioxidant and the protection of plants against abiotic stress [1,2,3]. However, the location of the plant, bee species, seasonality, climatic differences and solvent extraction processes can affect the chemical composition and biological properties [4,5,6].

The use of plant extracts as protection against skin photoaging is growing. Many extracts have components with photoprotective or synergistic activity in association with sunscreens, in addition to their high antioxidant potential. The use of natural extracts in cosmetic products, such as sunscreens for the prevention of skin diseases, has raised interest [7].

In the last few years, propolis has become the subject of intense pharmacological and chemical research. Across the world, propolis extract is suggested to improve health and prevent illnesses. Studies correlating the chemical composition with the biological activity are important, relating each type of propolis with its pharmacological application. The polyphenols and flavonoids present in propolis are important and they are presented by an absorption spectrum and can be used filter UV radiations, thus reducing the penetration of the radiation into the skin and lowering inflammation, oxidative stress and DNA damaging effects [7]. They have anti-inflammatory and antioxidant actions, which inhibit or delay the harmful effects of free radicals produced by UV radiation (singlet oxygen and hydroxyl free-radicals) [8]. The search for sun protection has intensified as the harmful effects of the sun have become more known and publicized. Exposure of human skin to ultraviolet light may cause sunburn to cells, accelerate skin aging and induce skin cancer.

In the different types of processes applied in the world to obtain propolis extracts, ethanol is the first recommended solvent, due to its chemical affinity with the propolis. Solvents, such as water, methanol, chloroform and ethylic ether, can also be recommended for the extraction specific constituents of propolis [9,10]. Another advantage of ethanol extraction is that waxes and other organic residues are removed [11].

Although several studies have already been carried out on the ethanol extract of Brazilian red propolis (EEPV) for sun protection, to the best of our knowledge, there are no studies evaluating the cytotoxicity and microbial activity of these extracts in sunscreens, at a concentration that increases the sun protection of the studied formulation. Recalling the importance of the cytotoxic characterization of a product that will come into contact with the skin and the antimicrobial properties of this product, in this work these assays were carried out for a deeper understanding of the developed sunscreen. The biological results reported here relate to the cytotoxic profile of EEPV (in cells present in the epidermal leaflet and in cells of the immune system) and were obtained via cell viability assay by MTT. Additionally, the antibacterial activity of propolis was evaluated against Gram-positive bacteria, such as *Staphylococcus aureus* and *Staphylococcus epidermidis*, important pathogens in nosocomial infections associated with catheters, implants and prostheses. Additionally, the production of nitric oxide in macrophage cells was evaluated and we confirmed that the propolis extract proposed in this work presents important anti-inflammatory properties for the development of formulations for application on the skin.

## 2. Results and Discussion

### 2.1. Phytochemical Screening

Qualitative phytochemical screening was carried on the ethanol extracts to identify flavonoids, phenols/tannins, terpenoids and saponins (Table 1).

The experiments relied on the alterations in color that occurred upon the combination of the extract with established reagents, as a means of identifying secondary metabolites. A vibrant yellow hue denoted the existence of flavonoids, while tannins and phenols were indicated by a brownish green or blue–black tint. Positive detection of terpenoids was represented by a reddish-brown coloring at the interface. Finally, the observation of a consistent, enduring foam indicated the presence of saponins.

Ethanol was used in the extractions at room temperature and at a hot temperature, because it is a polar solvent. Therefore, it is expected to perfectly extract active compounds of different polarities. In addition, studies show that aqueous ethanol 70% is able to extract more compounds from propolis than 100% ethanol and aqueous ethanol 90%, and also extracts more than when water is used [12].

Although other articles involving the extraction of propolis with aqueous ethanol 70% show the presence of various compounds [12,13], in the conditions used in our work only flavonoids and terpenoids were detected.

### 2.2. In Vitro Determination of the Sun Protection Factor (SPF)

As shown in Figure 1, the absorption spectra of EEPV room temperature and EEPV heated in solution exhibited absorbance in the UVC, UVB and UVA region, with absorption maxima at around 230 nm (UVC) and 300 nm (UVB and UVA), recommending that they are potential candidates for UV photoprotection.

The results of the in vitro determination of the SPF values of EEPV at room temperature and EEPV heated in different concentrations are shown in Table 2. According to Brazilian regulations [14], for a product to be classified as a genuine sunscreen, it must possess a sun protection factor (SPF) of at least six. As a photoprotective agent, the unadulterated EEPV demonstrates its ability to safeguard against the harmful effects of UV radiation.

The results of the in vitro determination of SPF values of the EEPV room temperature and EEPV heated in the formulation with cream are shown in Table 3. It can be observed that the SPF value of the EEPV in the photoprotective formulation was greater than the SPF value of the positive control (Sunscreen UVA–UVB 5% gel with Pemulen TR-1^®^). Some studies with Brazilian red propolis ethanol extract incorporated into photoprotective formulations have documented an increase in the SPF in the range of 2.2 to 3.8. This evidence shows that we found significant increases in sun protection in our work [15,16]. The best results found by us may be related to the composition extracted from propolis, which may be due to the extraction method used, the fauna and flora of the region where the propolis was collected and the chemical interaction between the plant’s metabolites and the chemical constituents of the formulation [15].

Pharmaceutical products are advised by both the FDA (the Food and Drug Administration of the United States) and the European Union to contain ingredients with SPF values greater than 15 to ensure proper protection against harmful UV radiation [17]. Therefore, both of the extracts studied in this work can be incorporated into Polawax cream at a concentration of 0.10 mg/mL to increase its sun protection.

### 2.3. Evaluation of Antioxidant Activity

To determine the antioxidant potential of the extracts, the DPPH scavenging activity and ABTS methods were employed [18,19]. As shown in Figure 2, EEPV presented an antioxidant activity and the effective concentration for reducing 50% of DPPH of 17 ± 1 and 12 ± 1 μg/mL for extraction at room temperature and a hot temperature, respectively. The IC_50_ results of the ABTS scavenging activity were 4.7 ± 0.3 and 4.7 ± 0.1 μg/mL for extractions at room temperature and a hot temperature (Figure 3). The extracts under investigation demonstrated a potent antioxidant effect, as demonstrated by their capacity to reduce DPPH and ABTS even at low concentrations, as anticipated. Data from the literature revealed that other propolis samples might have significantly higher IC_50_, varying from 0.070 to 932.0 mg/mL [20,21]. It is plausible that the disparity in results could be attributed not only to differences in geographic origin but also discrepancies in the extraction methods employed. The significance of working with chemically characterized propolis samples is confirmed by these findings, as differences in chemical compositions may account for divergent outcomes [22].

### 2.4. Determination of Total Phenolics and Flavonoids

Studies show that both phenolic compounds and flavonoids play an important role in the antioxidant activity of Brazilian propolis extracts [21,22,23,24,25]. Phenolic antioxidants function as sequesters of free radicals and sometimes as chelates of metals, acting as much in the initiating stage as in the propagation of the oxidation process [26,27]. In addition, phenolic and flavonoid compounds are the main components responsible for the functional property of propolis. In the current study, the total phenolics content of the EEPV room temperature and EEPV heated were, respectively, 10.6 ± 0.1 and 11.4 ± 0.1 mg GA/g of propolis, while the total flavonoids content from the same extracts (room and hot temperature) were, respectively, 10.9 ± 0.1 and 9.8 ± 0.1 mg QE/g of extract. Several investigations examining red propolis extracts have reported varying concentrations of phenolic and flavonoid compounds. Phenolic concentrations have been reported to range from 2.6 to 416 mg GAE/g propolis in certain studies [29], while flavonoid concentrations have been observed to vary between 6 and 43 mg QE/g propolis [28,30]. As previously mentioned, chemical variations in propolis samples are commonplace [4,5]. Additionally, the chemical composition of propolis extract can be influenced by the extraction method and solvent employed [6].

Our study yielded comparable values for total flavonoids and phenolics among the extracts analyzed. The antioxidant activity of the extracts also exhibited similar results.

### 2.5. Cell Viability

#### 2.5.1. Macrophage Cells

Cytotoxicity studies are essential to understand toxic effects potentially caused by the various substances in EEPV and establish safe doses for a given application. Here, we focused on the viability of macrophage cultures exposed to different EEPV doses (Figure 4). The results were compared with ISO10993-5, which considers a substance as cytotoxic when cell viability is less than 70% [31]. Thus, it was observed that the EEPV heated did not show cytotoxicity in any of the tested concentrations after 24 h of exposure, unlike the EEPV room temperature, which showed cytotoxicity at the highest concentration (500 μg/mL) and no cytotoxicity at the lowest concentrations (250 and 125 μg/mL). After establishing a safe EEPV dose, we proceeded to measuring the nitric oxide levels of the resulting macrophage culture.

#### 2.5.2. HaCaT Cells

HaCaT cells were evaluated by the MTT assay and were selected for this study because they compose the epidermal leaflet and are commonly used in studies that aim to analyze impacts on skin toxicity, skin barrier homeostasis and antioxidant activity [32]. Figure 5 shows the results of the MTT assay for the samples after 24, 48 and 72 h of exposure to the EEPV room temperature and EEPV heated suspensions at different concentrations (10, 25 and 50 μg/mL). The results indicate that cell viability was maintained at high levels, above 70% for all tested concentrations of EEPV room temperature and EEPV heated when compared with the three times analyzed cytotoxicity control shown in Figure 5a–c. A more significant drop in cell viability was observed for the groups exposed to 50 μg/mL after 48 and 72 h EEPV room temperature and EEPV heated suspension when compared with the viability control. However, the data showed values greater than 70% of cell viability. Karapetsas et al. (2019) [33] evaluated the effect of propolis extract obtained from different regions in Greece on the viability of HaCaT cells and determined average non-cytotoxic doses ranging from 26 to 28 µg/mL in a 24 h cell viability assay. Another study carried out in L-929 cells, a mouse fibroblast lineage, demonstrated cytotoxicity at doses greater than 1 µg/mL [34]. In contrast, the red propolis extracts obtained in this study showed significant cell viability using higher doses (50 µg/mL) at earlier time intervals [33].

### 2.6. Nitric Oxide

Nitric oxide (NO) has pro-inflammatory effects such as cytotoxicity, and mediation of cytokine-dependent processes that can lead to tissue injury and destruction. Thus, macrophages were stimulated with pro-inflammatory cytokines for the indirect assessment of NO levels and consequently their inflammation. Untreated macrophages stimulated or not with LPS + IFN-γ were used as controls. Figure 6 shows nitrite levels after treatment with the extracts. Significant differences were observed between the unstimulated and untreated control with LPS + IFN-y and the EEPV heated at the lowest concentration (125 μg/mL), and the EEPV at room temperature at the concentrations 250 and 125 μg/mL. Statistical differences were observed between untreated and LPS-stimulated controls + IFN-y of heated EEPV at the highest concentration (500 μg/mL), hence suggesting the potential anti-inflammatory effect of the extract at this concentration. Significant differences were also observed between the two controls with EEPV heated at an intermediate concentration (250 μg/mL) and EEPV at room temperature at its highest concentration (500 μg/mL). It should be noted that these results are very promising, since NO can even interfere with wound healing and inflammation caused by the sun. Human skin contains precursors of NO that can accumulate and move into the bloodstream after exposure to ultraviolet light, leading to inflammatory processes including systemic ones, which reinforces the importance of reducing NO production [35].

### 2.7. Antimicrobial Activity

Evaluations of the antimicrobial activity of EEPV room temperature and EEPV heated were performed using techniques that determined the minimum inhibitory concentration (MIC) and minimum bactericidal concentration (MBC) for the two Gram-positive bacteria defined for this study, *Staphylococcus aureus* and *Staphylococcus epidermidis*. Results are summarized in Table 4. As shown, the MIC of EEPV room temperature was 1000 μg/mL for both *S. aureus* and *S. epidermidis* and there was no MBC. On the other hand, MIC and MBC results for EEPV heated in *S. aureus* were 250 μg/mL and 500 μg/mL, respectively, and *S. epidermidis* presented an MIC of 500 μg/mL. Therefore, EEPV heated showed a more inhibitory effect on the growth of the evaluated bacteria, in addition to being able to eliminate *S. aureus* (MBC: 500 μg/mL). The geographic region of collection, as well as the varieties of bees, can interfere with the performance of the antibacterial activity of propolis extracts [36]. Despite this, Popova et al. (2017) demonstrated MIC values of 13 mg/mL for *S. aureus* and 12 mg/mL for *S. epidermidis* in a study carried out using variations of propolis from Poland, similar to the values found in our study for the EEPV room temperature [34].

### 2.8. Analysis of Chemical Components in the Ethanolic Extract of Red Propolis by UPLC-QTOF-MS/MS

The UPLC-QTOF-MS/MS analysis allowed the annotation of 40 substances for EEPV heated and 42 for EEPV room temperature and their MS^2^ spectra can be verified in the Appendix A. As can be seen in Table 5, both extracts have flavonoids and terpenoids in their compositions, which is in agreement with the data found in the qualitative phytochemical screening. Among them, the compounds Liquiritigenin, Isoliquiritigenin, Pinocembrin, Formononetin, Naringenin, Vestitol, Biochanin A and Daidzein have already been reported in samples of red propolis by other authors [37,38,39,40].

## 3. Materials and Methods

### 3.1. General Considerations

Solvents and reagents were purchased from Synth (Diadema, SP), Vetec (Duque de Caxias, RJ) and Neon (Suzano, SP), and used without further purification. The in vitro solar protection factor (SPF) was determined by the spectrophotometric method developed by Mansur [41,42]. Absorbance readings were performed on a Genesys 105 UV-VIS spectrophotometer equipped with 1 cm quartz cell and concentration range between 0.02 and 0.1 mg/mL. The cream sunscreen Polawax (Deionized Water-phase B, Germall 115-phase, Caprylic acid Caprylic Triglicer phase A, EDTA-phase B, Nipagim-phase B, Nipazol-Phase A, Polawax-phase A, Propyleneglicol-Phase B and BHT-phase A) was obtained from the NatureDerme Manipulation Pharmacy in Belo Horizonte, MG.

### 3.2. Plant Material

The crude samples of red propolis were bought in PharmaNéctar and obtained in Marechal Deodoro, State of Alagoas, located in the Northeastern Region of Brazil (SL 094237 and WL 355342).

### 3.3. Extraction of Ethanolic Extracts of Red Propolis-EEPV Heated

The crude sample of red propolis (2.0 g) was extracted with 15 mL of aqueous ethanol 70%, in a water bath at 70 °C for 30 min. After that, the sample was filtered on filter paper and 100 mL of aqueous ethanol 70% was added to the residue, and another alcoholic extraction was performed. The solution obtained from the extraction was dried and stored [18,19].

### 3.4. Extraction of Ethanolic Extracts of Red Propolis-EEPV Room Temperature

The crude sample of red propolis (2.0 g) was extracted with 15 mL of aqueous ethanol 70% for 48 h at room temperature and the resulting alcoholic extract was filtered under vacuum on filter paper and 100 mL of aqueous ethanol 70% was added to the residue, and another alcoholic extraction was completed. The solution obtained from the extraction was dried and stored [18,19].

### 3.5. Phytochemical Screening

We performed a qualitative phytochemical screening on EEPV using standard procedures with slight modifications to identify its phytoconstituents, including flavonoids, phenols/tannins, saponins and terpenoids. The screening was conducted on both the aqueous ethanol 70% EEPV at room temperature and the 70% EEPV at a hot temperature.

#### 3.5.1. Test for Flavonoids (Alkaline Reagent Test) [43,44]

To detect the presence of flavonoids, we dissolved 10 mg of the dry crude extract in 2 mL of 2% sodium hydroxide (NaOH) and observed the resulting color change. The development of a vivid yellow color indicated the presence of flavonoids.

#### 3.5.2. Test for Phenols/Tannins (Ferric Chloride Test) [43,44]

To detect the presence of tannins and phenols, we stirred 10 mg of the crude extract with 2 mL of distilled water, filtered the mixture and added a few drops of 2% ferric chloride (FeCl_3_). We then observed the resulting color change, and the appearance of a brownish green or blue-black color indicated the presence of tannins and phenols.

#### 3.5.3. Test for Saponins (Froth Test) [43,44]

Saponins are identified if, when vigorously shaking 5 mL of distilled water with 10 mg of ethanolic extract, there is the formation of a stable foam.

#### 3.5.4. Test for Terpenoids (Salkowski Test) [43,44,45]

To identify the presence of terpenoids, 10 mg of the dry crude extract was dissolved in 2 mL of chloroform (CHCl_3_), and 3 mL of concentrated sulfuric acid (H_2_SO_4_) was added to form a distinct layer. The presence of terpenoids was confirmed by the formation of a reddish-brown coloration at the interface.

### 3.6. Photoprotective Formulation of Red Propolis Ethanolic Extracts 70% (Room Temperature and Hot Temperature) in Polawax Cream

Dry EEPV 70% (room and hot temperature) was incorporated in the Polawax cream. The extracts were solubilized in ethanol propylene glycol 1:1 and incorporated in Polawax cream under agitation by 15 min. The final composition of formulations was 1% red propolis extract, 10% ethanol, 10% propylene glycol and creams fsf 100% [18,19].

### 3.7. In Vitro Determination of the Sun Protection Factor (SPF)

Dry EEPV 70% (room and hot temperature) were dissolved in ethanol (1 mg/mL) and diluted until obtaining concentrations of 0.020, 0.030, 0.050, 0.070 and 0.1 mg/mL. For the Polawax cream incorporated or not with EEPVs 70%, each formulation was weighed and the dilutions were performed in mixture ethanol/water 1:1, until obtaining a concentration of 0.01 g/mL. The in vitro SPF was determined for each concentration by the spectrophotometric method developed by Mansur [41,42]. Through the Mansur method equation, it was possible to determine the value of the SPF for this concentration [42]. The absorption readings were taken between 290 and 320 nm (UVB region). The experiment was performed in triplicate.

### 3.8. Evaluation of Antioxidant Activity

To evaluate the scavenging activity of the extract on DPPH free radicals, a modified method based on Almeida et al. [19] was used. Initially, the extracts were dissolved in ethanol to obtain stock solutions of 200 µg/mL. From these stock solutions, different aliquots were taken to obtain final solutions ranging from 2 to 117 µg/mL. Then, 1250 µL of 0.008% *w*/*v* DPPH solution in ethanol was added to each sample, and the final volume was adjusted to 3000 µL with ethanol. The mixture was vigorously shaken and left to stand in the dark at room temperature for 30 min. A negative control was prepared by mixing 1250 µL of DPPH with 2750 µL of ethanol and this was used to calculate the percentage inhibition of free radicals. Thereafter, the absorbance of the assay mixture was measured at 518 nm. DPPH radical scavenging activity was calculated using the equation:(1)$$I\%=\;\frac{Abs_{\mathrm{control}}-Abs_{\mathrm{sample}}}{Abs_{\mathrm{control}}}\;\times \;100$$

The concentration required to obtain a 50% antioxidant effect (EC_50_) was calculated by linear regression for the extracts.

The ABTS assay for decolorization of the ABTS cationic radical (ABTS˙^+^) was performed following the method described by Re et al. [46]. To prepare the working solution, ABTS (7.4 mmol/L) was mixed with potassium persulfate (2.6 mmol/L) and incubated at room temperature for 12–16 h in the dark. On the day of analysis, the solution was diluted with ethanol to obtain an absorbance of 0.70 (±0.02) at 650 nm. Extract concentrations were prepared as described previously. Next, 1.6 mL of the ABTS solution was added to the samples, and the final volume was adjusted to 2.0 mL with ethanol. The negative control was prepared by mixing 1.6 mL of the ABTS solution with 0.4 mL of ethanol. All samples were incubated for 30 min at room temperature (25 ± 2 °C) in the dark, and the absorbance was measured at 734 nm. The scavenging percentage of the sample and EC_50_ values were calculated as described previously.

### 3.9. Determination of Total Phenolics and Flavonoids

The total phenolic content was determined using the Folin–Ciocalteu method with slight modifications [47]. Briefly, 9.8 mg of samples were dissolved in 50 mL absolute ethanol and 1.6 mL of this solution was mixed with 1.2 mL of deionized water and 0.2 mL of Folin–Ciocalteu reagent (Cromoline). The mixture was agitated for 1 min, followed by the addition of 0.8 mL of sodium carbonate solution (7.5% *w*/*v*). After agitating for 30 s, 0.2 mL of water was added and the mixture was incubated for 2 h. The absorbance of the reaction mixture was measured at 725 nm against a deionized water blank using a spectrophotometer. Gallic acid (GA) was used as the standard. A standard calibration plot was generated at 725 nm using known concentrations of GA (3.24–12.96 μg/mL; r^2^ = 0.9964; y = 0.0858x + 0.0142). The concentrations of phenols in the test samples were calculated from the calibration plot and expressed as mg GA equivalent of phenol/g of sample.

The total flavonoid content was determined using the colorimetric method with aluminum chloride (AlCl_3_) according to the protocol described by Dowd et al. [48]. A 1.0 mL aliquot of sample solution (9.8 mg/mL in absolute ethanol) was mixed with 1.0 mL of 2% AlCl_3_. After incubation at room temperature for 10 min, the absorbance of the reaction mixture was measured at 420 nm against a blank consisting of ethanol and AlCl_3_ using a spectrophotometer. Quercetin (QE) was chosen as a standard. The total flavonoids were quantified by using a standard calibration curve of quercetin (2.0–20.0 μg/mL; r^2^ = 0.9979; y = 0.0739x − 0.0169). The experiment was performed in triplicate and the results were expressed as mg of quercetin equivalents per g of sample.

### 3.10. Biological Assays

#### 3.10.1. Sample Preparation

The EEPV room temperature and EEPV heated were subjected to UV radiation for 30 min for complete sterilization and, after that, the suspensions were prepared suspended in 2% DMSO (Dimethyl sulfoxide, Sigma-Aldrich^®^, St. Louis, MO, USA), homogenized for 1 min to prepare EEPVs suspension used in the biological assays according to pre-defined concentrations to perform in vitro studies.

#### 3.10.2. In Vitro Assays with Cell Culture

##### Cell Culture of RAW 264.7

The murine macrophage cell line RAW 264.7 was grown in basal culture medium containing RPMI Sigma-Aldrich^®^, supplemented with 10% FBS (Fetal Bovine Serum) (Gibco^®^, Waltham, MA, USA) and gentamicin (Thermo Fisher Scientific^®^, Waltham, MA, USA). Cells were incubated in a humidified atmosphere of 5% CO_2_ at 37 °C. Subcultures were performed at a 1:3 ratio and the culture medium was renewed every 2 to 3 days.

##### Cell Viability in RAW 264.7

Macrophages RAW 264.7, cultivated in RPMI 1640 medium (Sigma-Aldrich^®^), were distributed in a 96-well microtiter plate using a density of 5 × 10 ^5^ cell/well and after, they were incubated at 37 °C with 5% of CO_2_ for 24 h. The cells were treated with the samples dissolved in RPMI with dimethyl sulfoxide 2% (DMSO) at concentrations of 125, 250 and 500 µg/mL. Cell viability was evaluated using the 3-4,5-dimethyl-thiazol-2-yl-2,5-diphenyltetrazolium bromide (MTT) method [49]. The medium was removed and the wells washed with RPMI. Then, 100 µL of RPMI without phenol red containing 10% fetal bovine serum and 50 µL of filtered 2 mg/mL MTT was added to the wells. The plates were covered and incubated for 4 h. After this time, the reaction was stopped using 100 µL of DMSO and the absorbance of the samples was read in a microplate reader (570 nm). Percentage of cell viability was determined using the GraphPad Prism 8.0.1 software.

##### Nitric Oxide in RAW 264.7

Macrophages RAW 264.7 were distributed in a 96-well plate (5 × 10 ^5^ cell/well) and were incubated at 37 °C with 5% of CO_2_ for 24 h. After, they were treated with the samples dissolved in 2% DMSO (125, 250 and 125 µg/mL) and stimulated or not with LPS (10 μg/mL) and IFN-γ (100 ng/mL). After incubation for 24 h, the supernatant was removed and stored at −80 °C for analysis of nitric oxide. The experiment was performed in triplicate. Nitric oxide was analyzed indirectly by the quantification of nitrite by the Griess reaction method [50]. Nitrite concentrations were determined by extrapolation from the standard curve, constructed using various concentrations of sodium nitrite and the results were expressed as nanomolar (nmol/L). Absorbance values were measured using a microplate reader at 570 nm. Statistical differences were evaluated using the GraphPad Prism 8.0.1 software.

##### Cell Culture of HaCaT Cells

The human immortalized keratinocyte (HaCaT) cell line was obtained from Dr. Ivana Márcia Alves Diniz (UFMG). The cell line was grown in a basal culture medium containing a DMEM (Dulbecco’s Modified Eagle Medium) media (Gibco^®^), supplemented with 10% FBS (Fetal Bovine Serum) (Gibco^®^) and streptomycin antibiotics (100 μg/mL)/penicillin (500 U/mL) (Invitrogen^®^). Cells were incubated in a humidified atmosphere of 5% CO_2_ at 37 °C until 90% confluence and were detached from the plate for experiments using trypsin-EDTA (0.25%; Gibco^®^).

##### Cell Viability in HaCaT Cells

The biocompatibility of EEPV room temperature and EEPV heated was assessed via (4,5–dimethylthiazol-2-yl)-2,5–diphenyl-2H-tetrazolium bromide (MTT) assay (Invitrogen^®^), as described in [51]. HaCaT cells were seeded in a 48-well plate. After being cultivated for 24 h, the cells were treated with EEPV suspension (10, 25 and 50 µg/mL concentrations). The viability control received only 1x PBS and the cytotoxicity control was exposed 0.05% *v*/*v* Triton™ X-100 (Sigma-Aldrich^®^) per 15 min. After treatment, cell groups were evaluated at the end of three times (24, 48 and 72 h). The medium was removed and a solution containing 130 µL DMEM and 100 µL MTT (5 mg/mL) was added to each well. After 2 h, formazan crystals were observed under an optical microscope and dissolved in 130 µL 10% SDS in 0.01 mol/L HCl (Sigma-Aldrich^®^). For all the steps described above, culture plates were incubated at 37 °C in a humidified atmosphere of 5% CO_2_. After 18 h, 100 µL of the solution was transferred to a 96-well plate to measure the absorbance at 595 nm. The experiments were performed in biological triplicates.

#### 3.10.3. Antimicrobial Activity

##### Micro-Organisms and Culture Conditions

Microbiological assays using Gram-positive bacteria *Staphylococcus aureus* (ATCC 33591) and *Staphylococcus epidermidis*, obtained by Dr. Lirlândia Pires de Sousa (UFMG), were performed. The antibacterial activity was determined using the minimum inhibitory concentration (MIC) for susceptible bacteria. The bacteria were activated in liquid brain–heart infusion medium (BHI) (Sigma-Aldrich^®^) at 35 °C for 24 h. After, they were seeded, through compound streaking in a Petri dish containing Nutrient Agar and incubated for 24 h at 35 °C. Primary colonies were collected and examined in a test tube containing 10 mL of sterile 0.145 mol/L saline solution (8.5 g/L of NaCl; 0.85% saline) until reaching turbidity equivalent to a 0.5 McFarland scale (in the case of bacteria it corresponds to 10^8^ CFU/mL).

##### Minimum Inhibitory Concentration (MIC) and Minimum Bactericidal Concentration (MBC)

The MIC and MBC of EEPV room temperature and EEPV heated were determined using the broth microdilution method as described by the Clinical and Laboratory Standards Institute (CLSI, 2003) [52], with modifications. For MIC assays, S. aureus and S. epidermidis bacteria were grown in Mueller Hinton medium (MHB; Merck^®^, Darmstadt, Germany), with pH adjusted between 7.2 and 7.4 at room temperature of 25 °C. Initially, a 50μL aliquot of the previously homogenized pre-inoculum was added to 10 mL of MHB for making the inoculum, resulting in a final concentration of 10^5^ CFU/mL in the wells. The stock solution of EEPV room temperature and EEPV heated were prepared at a concentration of 20,000 µg/mL. From this concentration, the dilution was performed directly in the wells and, subsequently, a 1:10 dilution was performed in the first well and, from the second, a dilution in a series of 1:2. With the serial dilution of the antimicrobial agent concluded, 100 µL of the inoculum was added to the wells resulting in a new dilution of 1:2 and a final volume of 200 µL in the wells. Therefore, for the stock solution at a concentration of 20,000 µg/mL, a concentration of 1000 µg/mL was obtained in the first well. Sequentially, in the other wells, concentrations of 500 µg/mL, 250 µg/mL, 125 µg/mL, 62.5 µg/mL, 31.25 µg/mL, 15.63 µg/mL and 7.81 µg/mL were obtained. The middle blank and growth control were included in all 96-well plates used, serving as a reference for MIC determination.

The blank of the sample was performed for each agent tested, and the positive and negative controls were performed only once for each microorganism. The experiments were performed in triplicates. The results were read visually after 18 h of incubation at 35 °C. The MIC was considered as the lowest concentration of the tested agent capable of preventing 100% visible microbial growth. For MBC assays, a volume of 20 μL of each concentration was inoculated into petri dishes containing Mueller–Hinton agar. These plates were incubated 24 h/37 °C. After this period, the plates were read.

#### 3.10.4. Statistical Analysis

The cell viability of HaCaT cells was analyzed using statistical methods through one-way ANOVA, followed by Tukey’s post-test. The percentage values were reported as mean ± SEM using the GraphPad Prism 6 software. Statistical significance was considered when *p* ≤ 0.05. On the other hand, RAW 264.7 cell assays were analyzed using one-way ANOVA followed by Dunnett’s post-test. The results were reported as mean ± SD, and the GraphPad Prism 8 software was utilized. Differences were considered significant when *p* ≤ 0.05 [53].

### 3.11. Analysis of Chemical Components in the Ethanolic Extract of red Propolis by UPLC-QTOF-MS/MS

The chemical profile analysis of red propolis extract was performed by liquid chromatography coupled to mass spectrometry (LC-MS) according to Azevedo et al., 2022. Both samples were solubilized in acetonitrile at 200 ppm and the chromatographic separation occurred on a C18 column (Zorbax Eclipse). Briefly, the chromatographic conditions were: flow rate of 0.350 mL/min for eluents A (H_2_O acidified with 0.1% formic acid) and B (acetonitrile acidified with 0.1% formic acid) in gradient mode. TOF-MS analyses were performed at three power levels for positive and negative mode. The annotation of the compounds was performed using the GNPS (Global Natural Products Network) platform and the MS-Finder and MS-Dial 4.60 software.

## 4. Conclusions

The results indicate that the ethanolic red propolis extracts possess photoprotective properties, as observed in the assays when the extract was incorporated into the cream sunscreen Polawax. As far as we know, this is the first study on the photoprotective effect of ethanolic extracts from red propolis incorporated into a formulation containing cream sunscreen Polawax. Although many authors show a correlation between phenol content, flavonoid content and antioxidant activity in the value of solar protection, our work concludes differently. The extraction method used was not sufficient to remove many phenols and flavonoids, but the antioxidant activity was high. Thus, we correlated the high value of sun protection found only with antioxidant activity. In addition, the results showed that the incorporation of ethanol extract of propolis in photoprotective formulations, in addition to intensifying the values of sun protection, ensured, for the preparation, the other properties of propolis, such as antimicrobial, anti-inflammatory, antioxidant and healing. Therefore, this study suggests that it is a promising source of natural compounds for the development of new photoprotective formulations.

## Figures and Tables

**Figure 1 ijms-24-05112-f001:**
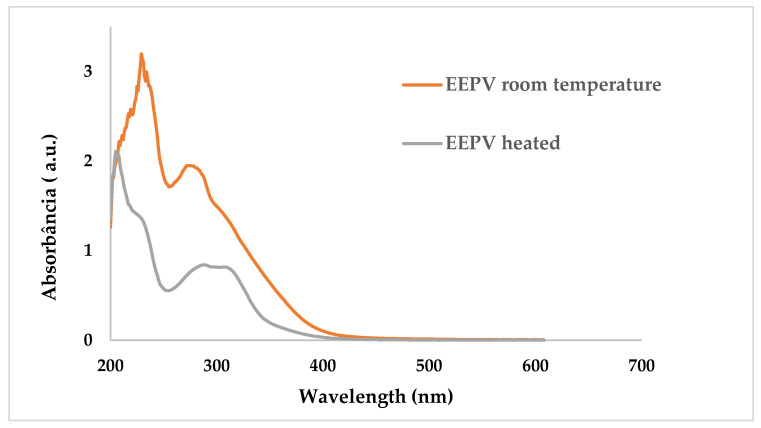
UV/Vis absorption spectra of EEPV 70% prepared heated in solution and EEPV 70% prepared at room temperature.

**Figure 2 ijms-24-05112-f002:**
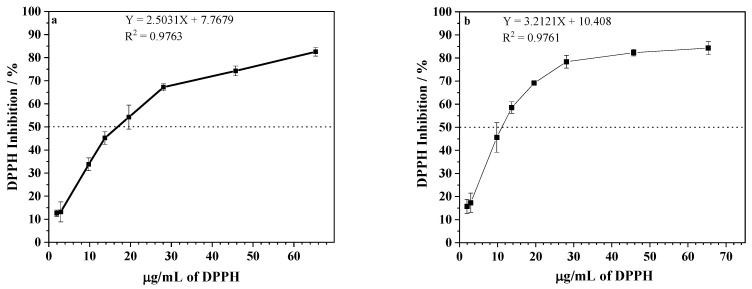
DPPH inhibition (%) after incubation with different propolis concentrations of EEPV (2.0–65.3 μg/mL) extracted at: (**a**) room temperature, and (**b**) a temperature of 70 °C. Data represent mean ± SD of 3 independent assays in duplicate. The dashed line indicates the concentration of propolis that inhibited 50% of DPPH. IC_50_ was calculated from the calibration curve determined by linear regression.

**Figure 3 ijms-24-05112-f003:**
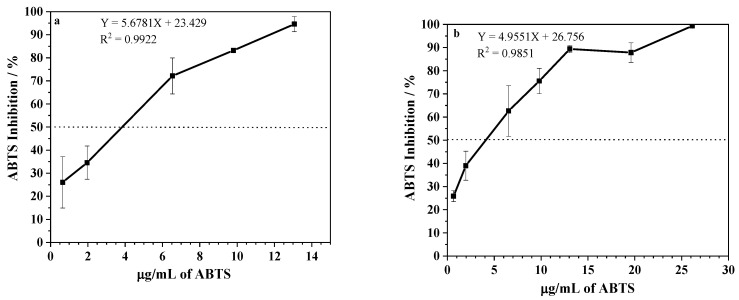
ABTS inhibition (%) after incubation with different concentrations of EEPV (0.6–26 μg/mL) extracted at: (**a**) room temperature, and (**b**) a temperature of 70 °C. Data represent mean ± SD of 3 independent assays. The dashed line indicates the concentration of propolis that inhibited 50% of DPPH. IC_50_ was calculated from the calibration curve determined by linear regression.

**Figure 4 ijms-24-05112-f004:**
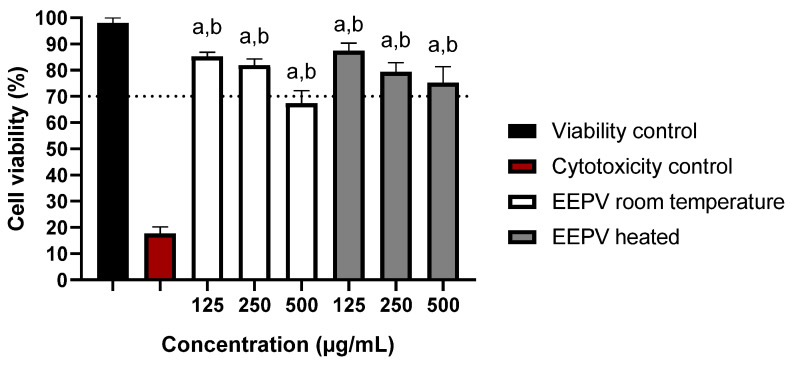
Cell viability in RAW 264.7 cells exposed to different concentrations of EEPV room temperature and EEPV heated (125, 250 and 500 μg/mL) at 24 h. Results represent the mean ± SD of triplicates of the experiments; (a) denotes a significant difference compared to the viability control (*p* ≤ 0.05); (b) denotes a significant difference in relation to the cytotoxicity control (DMSO 50%) (*p* ≤ 0.05), as determined by one-way ANOVA followed by a Dunnett’s post-test.

**Figure 5 ijms-24-05112-f005:**
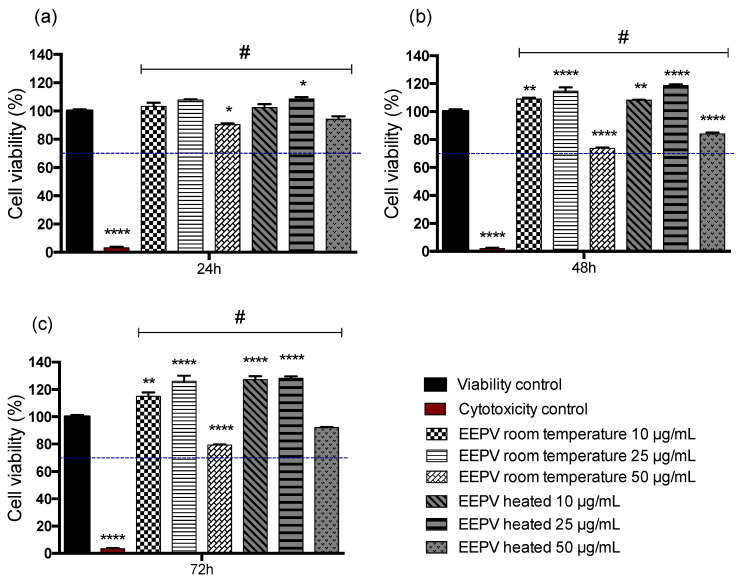
The MTT assay was employed to evaluate the cell viability of HaCaT cells exposed to different concentrations (10, 25 and 50 μg/mL) of room temperature and heated suspensions of EEPV at (**a**) 24, (**b**) 48 and (**c**) 72 h. The mean cell viability was normalized to that of the control group, which was exposed to PBS. A cytotoxicity control group exposed to 0.05% *v*/*v* Triton™ X-100 was also used. The error bars represent ± SEM, and statistical significance was indicated by * for *p* ≤ 0.05, ** for *p* ≤ 0.01, and **** for *p* ≤ 0.0001 compared to the Viability control. In addition, # was used for *p* < 0.0001 compared to the Cytotoxicity control. The statistical analysis was conducted using one-way ANOVA followed by a Tukey post-test.

**Figure 6 ijms-24-05112-f006:**
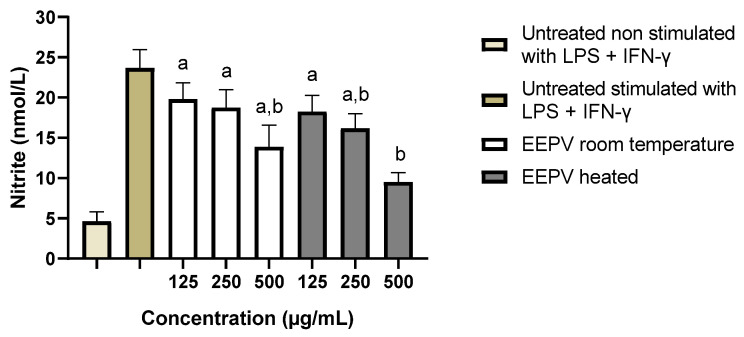
Nitric oxide (NO) production in RAW 264.7 cells after exposure to three different doses of EEPV at room temperature and EEPV heated (125, 250 and 500 μg/mL) for 24 h. The following controls were included in this assay: (positive control) untreated and non-stimulated with LPS + IFN-γ, (negative control) untreated macrophages stimulated with LPS + IFN-γ. Each bar shows the mean ± SD of triplicates of experiments. (a) denotes a significant difference in relation to the untreated control and not stimulated with LPS + IFN-y (*p* ≤ 0.05); (b) denotes a significant difference compared to untreated control and stimulated with LPS + IFN-y (*p* ≤ 0.05), as determined by one-way ANOVA followed by a Dunnett’s post-test.

**Table 1 ijms-24-05112-t001:** Phytochemical analysis of the ethanolic red propolis extracts (EEPV).

Phytochemicals	EEPV Room Temperature	EEPV Heated
Flavonoids (alkaline reagent test)	(+)	(+)
Phenols/Tannins (ferric chloride test)	(−)	(−)
Terpenoids (Salkowski test)	(+)	(+)
Saponins (froth test)	(−)	(−)

**Table 2 ijms-24-05112-t002:** SPF values (mean ± SD) calculated from EEPV in different concentrations.

Concentration mg/mL	SPF	
	EPPV Room Temperature	EEPV Heated
0.02	4.63955 ± 0.0194300	0.273258 ± 0.003261
0.03	5.400514 ± 0.677986	0.279551 ± 0.009742
0.05	8.278276 ± 0.021676	4.576193 ± 0.005881
0.07	9.214948 ± 1.636204	5.892001 ± 0.001900
0.10	10.77066 ± 0.215202	8.067012 ± 0.007059

**Table 3 ijms-24-05112-t003:** SPF values calculated from Polawax cream (negative control), Polawax cream incorporated with EEP (room temperature and heated), and Sunscreen UVA-UVB 5% gel with Pemulen TR-1^®^ (positive control).

Formulations in the Concentration of 0.10 mg/mL	SPF
Polawax cream (negative control)	10.7705 ± 0.5311
EPPV room temperature with Polawax cream	38.0588 ± 1.0661
EPPV heated with Polawax cream	42.8509 ± 0.8840
Sunscreen UVA-UVB 5% gel with Pemulen TR-1^®^ (positive control)	28.9694 ± 1.2315

**Table 4 ijms-24-05112-t004:** Antibacterial activity of the ethanolic red propolis extracts (EPPV).

Strain	EEPV Room Temperature	EEPV Heated
	MIC (MBC) (μg/mL)	
**Gram (+) Bacteria**		
*S. aureus*	1000 (-)	250 (500)
*S. epidermidis*	1000 (-)	500 (-)

MIC—Minimum Inhibitory Concentration; MBC—Minimum Bactericidal Concentration.

**Table 5 ijms-24-05112-t005:** Detected compounds from ethanolic extract of red propolis by UPLC-QTOF-MS/MS.

Nº	tR (min)	Precursor Ion (*m*/*z*)	Molecular Formula	Adduct	Error (ppm)	Ions-Fragment (*m/z*)	Annotation	Class
1	5.033	177.019	C_9_H_6_O_4_	[M-H]^−^	−0.09	149.0230; 121..0294;92.0264; 77.0398	Daphnetin	Coumarin
2	7.123	419.135	C_20_H_22_O_7_	[M+FA-H]^−^	−0.11	373.1279; 327.1226; 208.0727; 151.0394	Wikstromol	Lignan
3	7.187	235.169	C_15_H_22_O_2_	[M+H]^+^	2.57	189.1639; 133.1013; 105.0695; 93.0696	Valerenic acid *	Terpene
4	7.323	287.056	C_15_H_12_O_6_	[M-H]^−^	−0.13	259.0592; 177.0551; 125.0241; 83.0139	Dihydrokaempferol	Flavonoid
5	7.324	303.086	C_16_H_14_O_6_	[M+H]^+^	1.53	177.0537; 153.0543; 138.0313; 79.0546	Hesperetin	Flavonoid
6	7.525	359.149	C_20_H_24_O_6_	[M-H]^−^	0.18	329.1378; 192.0788; 178.0622; 160.0530	Lariciresinol	Lignan
7	7.999	253.05	C_15_H_10_O_4_	[M-H]^−^	−0.85	224.0473; 209.0597; 135.0086; 117.0340	Daidzein	Flavonoid
8	8.27	255.066	C_15_H_12_O_4_	[M-H]^−^	−0.26	135.0086; 119.0500; 91.0189	DL-Liquiritigenin *	Flavonoid
9	8.404	283.061	C_16_H_12_O_5_	[M-H]^−^	0.17	268.0374; 240.0419; 211.0390; 184.0521	Biochanin A	Flavonoid
10	8.672	357.134	C_20_H_22_O_6_	[M-H]^−^	0.32	342.1098; 176.0474; 151.0396; 136.0159	Pinoresinol *	Lignan
11	8.674	313.0712	C_17_H_14_O_6_	[M-H]^−^	0.04	298.0474; 283.0243; 269.0443; 255.0293	Cirsimaritin	Flavonoid
12	9.012	255.066	C_15_H_12_O_4_	[M-H]^−^	−1.44	237.0553; 209.0604; 135.0085; 109.0294	Dihydrodaidzein	Flavonoid
13	9.144	327.087	C_18_H_16_O_6_	[M-H]^−^	−0.11	311.0581; 297.0400; 269.0418; 146.9379	Kaempferol-3,7,4′-trimethyl ether	Flavonoid
14	9.348	285.077	C_16_H_14_O_5_	[M-H]^−^	−0.71	270.0528; 149.9956; 124.0163; 109.0294	7-Hydroxy-6-methoxydihydroflavonol	Flavonoid
15	9.418	271.0606	C_15_H_12_O_5_	[M-H]^−^	0.18	151.0034; 119.0501; 107.0136; 83.0137	Naringenin	Flavonoid
16	9.481	297.0762	C_17_H_14_O_5_	[M-H]^−^	0.33	281.0447; 267.0293; 253.0504; 239.0343	2′-Methoxyformonetin	Flavonoid
17	9.481	359.149	C_20_H_22_O_6_	[M+H]^+^	1.85	177.0915; 137.0595; 131.0492; 74.0951	Matairesinol	Lignan
18	9.551	329.066	C_17_H_14_O_7_	[M-H]^−^	−0.52	314.0428; 299.0197; 271.0242; 161.0239	3,7-Dimethylquercetin	Flavonoid
19	9.617	285.076	C_16_H_12_O_5_	[M+H]^+^	1.75	229.0858; 215.0701; 187.0751; 151.0389	Glycitein	Flavonoid
20	10.16	253.0865	C_16_H_14_O_3_	[M-H]^−^	−0.12	238.0627; 255.0535; 210.0683	Dalbergichromene	Flavonoid
21	10.22	303.1226	C_17_H_18_O_5_	[M+H]^+^	2.14	285.0756; 167.0701; 123.0441; 107.0492	Isomucronulatol	Flavonoid
22	10.36	257.0810	C_15_H_12_O_4_	[M+H]^+^	1.49	211.0752; 147.0443; 137.0234; 119.0491	Liquiritigenin	Flavonoid
23	10.43	255.066	C_15_H_12_O_4_	[M-H]^−^	−0.26	135.0086; 119.0500; 91.0190	Isoliquiritigenin	Chalcone
24	10.56	301.071	C_16_H_12_O_6_	[M+H]^+^	2.04	286.0466; 269.0446; 241.0493; 153.0181	Chrysoeriol	Flavonoid
25	10.63	273.1126	C_16_H_16_O_4_	[M+H]^+^	0.31	163.0754; 149.0596; 137.0598; 123.0441	Isovestitol	Flavonoid
26	10.63	267.0658	C_16_H_12_O_4_	[M-H]^−^	−0.25	252.0422; 223.0395; 195.0445; 132.0212	Formononetin	Flavonoid
27	10.9	241.0861	C_15_H_12_O_3_	[M+H]^+^	1.53	195.0801; 137.0233; 131.0490; 103.0543	7-Hydroxyflavanone	Flavonoid
28	11.1	273.1126	C_16_H_16_O_4_	[M+H]^+^	0.31	163.0753; 149.0598; 137.0597; 123.0440	Vestitol	Flavonoid
29	11.17	271.096	C_16_H_14_O_4_	[M+H]^+^	2.34	161.0596; 137.0597; 123.0438; 109.0648	5-Hydroxy-7-methoxyflavanone	Flavonoid
30	11.57	235.169	C_15_H_22_O_2_	[M+H]^+^	2.57	189.1628; 133.1007; 119.0855; 107.0855	Curcumenol	Terpene
31	11.64	269.0814	C_16_H_14_O_4_	[M-H]^−^	−0.06	254.0580; 239.0345; 226.0626; 210.0679	Dalbergione, 4-Methoxy-4′-Hydroxy-	Neoflavonoid
32	11.84	255.102	C_16_H_14_O_3_	[M+H]^+^	2.04	161.0596; 151.0389; 131.0490; 107.0492	2′-Hydroxy-4′-Methoxychalcone	Chalcone
33	12.11	255.066	C_15_H_12_O_4_	[M-H]^−^	−0.26	213.0546; 171.0442; 151.0033; 107.0136	Pinocembrin	Flavonoid
34	12.38	283.061	C_16_H_12_O_5_	[M-H]^−^	0.17	268.0372; 239.0343; 224.0470; 132.0208	Acacetin	Flavonoid
35	13.26	241.0861	C_15_H_12_O_3_	[M+H]^+^	1.53	195.0798; 137.0232; 131.0488; 103.0541	2′,4′-Dihydroxychalcone	Chalcone
36	13.36	432.238	C_22_H_30_O_6_	[M+ACN+H]^+^	1.42	135.0802; 129.0543; 119.0856; 107.0856	7b,9-Dihydroxy-3-(hydroxymethyl)-1,1,6,8-tetramethyl-5-oxo-1,1a,1b,4,4a,5,7a,7b,8,9-decahydro-9aH-cyclopropa[3,4]benzo[1,2-e]azulen-9a-ylacetate	Terpene
37	14.41	241.087	C_15_H_14_O_3_	[M-H]^−^	−0.13	226.0619; 213.0904; 186.0321; 150.9153	Lapachol	Quinone
38	16.22	203.179	C_15_H_22_	[M+H]^+^	3.82	147.1165; 119.0845; 105.0696; 95.0856	Alpha-Curcumene	Terpene
39	17.57	205.195	C_15_H_26_O	[M+H-H2O]^+^	3.05	121.1006; 107.0854; 93.0699; 81.0701	Alpha-Bisabolol	Terpene
40	17.74	409.1652	C_24_H_26_O_6_	[M-H]^−^	−0.21	394.1407; 366.1466; 351.0859; 339.0851	Alpha-Mangostin	Xanthone
41	20.07	439.357	C_30_H_48_O_3_	[M+H-H2O]^+^	1.38	203.1787; 191.1791; 109.1014; 95.0856	Oleanolic acid	Terpene
42	20.14	311.1643	C_20_H_22_O_3_	[M+H]^+^	1.35	203.1062; 177.0542; 161.0960; 135.0438	Dihydrocordoin ^#^	Chalcone
43	21.42	413.269	C_26_H_38_O_4_	[M-H]^−^	−0.04	344.1981; 301.1437; 289.1435; 233.0815	Lupulone	Terpene

* Compound present only in the EEPV room temperature; ^#^ Compound present only in the EEPV heated.

## Data Availability

Not applicable.

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
