# Peer review of "Anti-Inflammatory, Antimicrobial, Antioxidant and Photoprotective Investigation of Red Propolis Extract as Sunscreen Formulation in Polawax Cream"

_ijms, 2023, doi:10.3390/ijms24065112_

Round 1

Reviewer 1 Report

The manuscript submitted by Viviane Martins Rebello dos Santos et al. describes a deep study on ethanolic extracts of red propolis, particularly focused on their evaluation as additives for sunscreen formulations.

Two slightly different extraction methods are presented, and for each one of the solutions recovered, a study on their anti-inflammatory, antimicrobial, antioxidant, and photoprotective properties were made.

In my opinion, the manuscript will be suitable for publication in IJMS after some minor revisions, in particular:

- The Supplementary material is cited in the text but it was not available for this review; please provide all the related files

- Since the main purpose of the paper is to study the ability of the extracts as additives in sunscreens, UV-Vis spectra of these extracts should be reported to characterize the optical properties of the different materials

- For sake of clarity, when the extractions with non-absolute ethanol are cited in the text, it should be specified that the solvent is aqueous ethanol (instead that only 70% ethanol)

- In the paper, a major problem for the clarity of the manuscript is represented by the discussion of experimental procedures without recalling the section of the manuscript in which there is a detailed description of them. Moreover, when the procedure/technique is well known, a reference should be always recalled. More in detail: Page 3, Section 2.1, Lines 104-109: recall the section of "Material and Methods" when describing the different tests, and add the corresponding references in the text; 2) Page 4, Section 2.3, Line 144: Add References; 3) Page 6, Section 2.5.2, Line 215: add References after "(2019)"; 4) Page 8, Section 2.6, Line 257: add References; Page 8, Section 2.7, Line 270: add References after "....propolis extracts.";  5) Page 9, Section 3.1, Line 291 add References after "....Mansur."; 6) Page 10, Sections 3.5.1 to 3.5.4: add the corresponding references for each test; 7) Page 14, Section 3.10.4, Line 499: add References after "...post-test."

- Page 4, Figure 1 and Page 5, Figure 2: in the caption of the figures, please specify the right temperature of extraction instead that employing the general term "hot"

- Page 8, Line 279: as already said, there is a recall of the supplementary material, but the file was not provided. Please add the corresponding file.

- Page 9, Line 292: add the concentration (or the concentration range) employed for the UV-Vis measurements

- Page 11, Section 3.9: why did you choose those particular standards for the different tests?

- Page 14, Section 3.11: when an ESI/MS instrument is employed, an important variable to characterize the experimental procedure is represented by the potentials applied to the injector, in order to be able to reproduce the ionization observed. Please add the corresponding information for your apparatus.

- The behavior of the two kinds of ethanolic extracts is not always the same in the tests studied, and the qualitative composition (as revealed by UPLC-QTOF-MS/MS) is almost te same. Considering these aspects, what could be a reasonable explanation for this difference? 

- An interesting aspect, in order to think about a final application of the ethanolic extracts could be represented by the efficiency of extraction, in terms of the ratio between the initial biomass and the extracted material. Could you provide this information? Could it be possible to compare it with other different studies?

- When considering the cell viability, the study was performed only on the extracts and not on the final formulations too. Is there a reason for this choice to not going further?

Author Response

Response to reviewer 1

- The Supplementary material is cited in the text but it was not available for this review; please provide all the related files

Response: Sorry for the glitch. Supplementary material has been added to the requested files.

- Since the main purpose of the paper is to study the ability of the extracts as additives in sunscreens, UV-Vis spectra of these should be reported to characterize the optical properties of the different materials

Response: UV-Vis spectra of these extracts have been added and reported.

The absorption spectra of EEPV at room temperature and EEPV heated exhibited absorbance in the UVC, UVB and UVA region, with absorption maxima at around 230 nm (UVC) and 300 nm (UVB and UVA), suggesting that are potential candidates for UV photoprotection.

As shown in Figure 1, the absorption spectra of EEPV room temperature and EEPV heated in solution exhibited absorbance in the UVC, UVB and UVA region, with absorption maxima at around 230 nm (UVC) and 300 nm (UVB and UVA), suggesting that are potential candidates for UV photoprotection.

Figure 1. UV/Vis absorption spectra of EEPV  70 % prepared at heated. and EEPV  70 % prepared at room temperature.

- For sake of clarity, when the extractions with non-absolute ethanol are cited in the text, it should be specified that the solvent is aqueous ethanol (instead that only 70% ethanol)

Response: Modified and specified as aqueous ethanol 70%

- In the paper, a major problem for the clarity of the manuscript is represented by the discussion of experimental procedures without recalling the section of the manuscript in which there is a detailed description of them. Moreover, when the procedure/technique is well known, a reference should be always recalled. More in detail: Page 3, Section 2.1, Lines 104-109: recall the section of "Material and Methods" when describing the different tests, and add the corresponding references in the text; 2) Page 4, Section 2.3, Line 144: Add References; 3) Page 6, Section 2.5.2, Line 215: add References after "(2019)"; 4) Page 8, Section 2.6, Line 257: add References; Page 8, Section 2.7, Line 270: add References after "....propolis extracts.";  5) Page 9, Section 3.1, Line 291 add References after "....Mansur."; 6) Page 10, Sections 3.5.1 to 3.5.4: add the corresponding references for each test; 7) Page 14, Section 3.10.4, Line 499: add References after "...post-test."

Page 3, Section 2.1, Lines 104-109: recall the section of "Material and Methods" when describing the different tests, and add the corresponding references in the text;

Response: The experiments relied on the alterations in color that occurred upon the combination of the extract with established reagents, as a means of identifying secondary metabolites. A vibrant yellow hue denoted the existence of flavonoids, while tannins and phenols were indicated by a brownish green or blue-black tint. Positive detection of terpenoids was represented by a reddish-brown coloring at the interface. Finally, the observation of a consistent, enduring foam indicated the presence of saponins.

 2) Page 4, Section 2.3, Line 144: Add References;

Response: References have been added

3) Page 6, Section 2.5.2, Line 215: add References after "(2019)";

Response: References have been added

4) Page 8, Section 2.6, Line 257: add References;

Response: Page 8, Section 2.6, Line 257 is legend of figure.

Page 8, Section 2.7, Line 270: add References after "....propolis extracts."; 

Response: References have been added

5) Page 9, Section 3.1, Line 291 add References after "....Mansur.";

Response: We included the references after Mansur.

 6) Page 10, Sections 3.5.1 to 3.5.4: add the corresponding references for each test;

Response: We included the references used in each of the tests.

 7) Page 14, Section 3.10.4, Line 499: add References after "...post-test."

Response: References have been added

- Page 4, Figure 1 and Page 5, Figure 2: in the caption of the figures, please specify the right temperature of extraction instead that employing the general term "hot"

Response: The right temperature was set and the word 'hot' was removed

- Page 8, Line 279: as already said, there is a recall of the supplementary material, but the file was not provided. Please add the corresponding file.

Response: Supplementary material has been added to the requested files.

- Page 9, Line 292: add the concentration (or the concentration range) employed for the UV-Vis measurements.

Response: the concentration range have been added

- Page 11, Section 3.9: why did you choose those particular standards for the different tests?

Response: In the references used, the authors refer to the compounds we used in our work. Thus, in the determination of total phenolics, Bonoli et al. [Bonoli, M.; Verardo, V.; Marconi, E.; Caboni, M.F. Antioxidant phenols in barley (Hordeum vulgare L.) flour: comparative spectrophotometric study among extraction methods of free and bound phenolic compounds. J. Agric. Food Chem. 2004, 52(16), 5195–5200] used gallic acid as standard. In the determination of flavonoids, Dowd (Dowd, L.E. Spectrophotometric Determination of Quercetin. Anal. Chem. 1959, 31, 1184-1187) used quercetin as the standard.

Other authors have also used these same patterns in the same tests:

- Gallic acid in the determination of total phenolics:

  1. i) W.A.S. da Almeida, A.S. Antunes, R.G. Penido, H.S.G. da Correa, A.M. do Nascimento, A.L. Andrade, V.R. Santos, T. Cazati, T.R. Amparo, G.H.B. de Souza, K.M. Freitas, O.D.H. dos Santos, L.R. Dutra Sousa, V.M.R. dos Santos. Photoprotective activity and increase of SPF in sunscreen formulation using lyophilized red propolis extracts from Alagoas. Revista Brasileira de Farmacognosia, 29:373–380 (2019)
  2. ii) J.B. Seibert, J.P. Bautista-Silva, T.R. Amparo, A. Petit, P. Pervier, J.C.S. Almeida, M.C. Azevedo, B.M. Silveira, G.C. Brandão, G.H.B. de Souza, L.F.M. Teixeira, O.D.H. dos Santos. Development of propolis nanoemulsion with antioxidant and antimicrobial activity for use as a potential natural preservative. Food Chemistry, 287:61-67(2019).

iii) A.L.I. Piyathunga, M.A.L.N. Mallawaarachchi, W.M.T. Madhujith. Phenolic Content and Antioxidant Capacity of Selected Underutilized Fruits Grown In Sri Lanka. Tropical Agricultural Research, 27(3):277-286(2016).

  1. iv) S. Finardi, T.G. Hoffmann, B.L. Angioletti, E. Mueller, R.S. Lazzaris, S.L. Bertoli, M. Hlebova¡, M. Khayrullin, N. Nikolaeva, M.A. Shariati, C.K. de Souza. Development and application of antioxidant coating on Fragaria stored under isothermal conditions. Journal of microbiology, biotechnology and food sciences. 11(4): e5432(2022).
  • Quercetin in the measurement of total flavonoids:
  1. A.S. da Almeida, A.S. Antunes, R.G. Penido, H.S.G. da Correa, A.M. do Nascimento, A.L. Andrade, V.R. Santos, T. Cazati, T.R. Amparo, G.H.B. de Souza, K.M. Freitas, O.D.H. dos Santos, L.R. Dutra Sousa, V.M.R. dos Santos. Photoprotective activity and increase of SPF in sunscreen formulation using lyophilized red propolis extracts from Alagoas. Revista Brasileira de Farmacognosia, 29:373–380(2019).
  2. ii) J.B. Seibert, J.P. Bautista-Silva, T.R. Amparo, A. Petit, P. Pervier, J.C.S. Almeida, M.C. Azevedo, B.M. Silveira, G.C. Brandão, G.H.B. de Souza, L.F.M. Teixeira, O.D.H. dos Santos. Development of propolis nanoemulsion with antioxidant and antimicrobial activity for use as a potential natural preservative. Food Chemistry, 287:61-67(2019).
  • Sulastri, M.S. Zubair, N.I. Anas, S. Abidin, R. Hardani, R. Yulianti, Aliyah. Total Phenolic, Total Flavonoid, Quercetin Content and Antioxidant Activity of Standardized Extract of Moringa oleifera Leaf from Regions with Different Elevation. Pharmacogn J, 10(6):Suppl:s104-s108(2018).
  1. Sambandam, D. Thiyagarajan, A. Ayyaswamy, P. Raman. Extraction and isolation of flavonoid quercetin from the leaves of Trigonella foenum-graecum and their anti-oxidant activity. International Journal of Pharmacy and Pharmaceutical Sciences, 8(6)(2016).

- Page 14, Section 3.11: when an ESI/MS instrument is employed, an important variable to characterize the experimental procedure is represented by the potentials applied to the injector, in order to be able to reproduce the ionization observed. Please add the corresponding information for your apparatus.

Response: We agree with the reviewer. However, as mentioned in the text, the methodology used to characterize the chemical profile of the extracts was the same as that of Azevedo et al. (2022), as can be seen in the excerpt highlighted below:

“The operational source parameters for TOF-MS mode were performed at three power levels: low (capillary voltage: 2200 V, fragmentation voltage: 110 V, nozzle voltage: 300 V), medium (capillary voltage: 2500 V, fragmentation voltage: 120 V, nozzle voltage: 500 V) and high (capillary voltage: 3000 V, fragmentation voltage: 130 V, nozzle voltage: 600 V). Auto MS/MS mode was performed using collision energy table with the following values: 150-500 Da (20-25 eV), 500-1000 Da (25-50eV) and 1000-1500 (50-60 eV).”

- The behavior of the two kinds of ethanolic extracts is not always the same in the tests studied, and the qualitative composition (as revealed by UPLC-QTOF-MS/MS) is almost te same. Considering these aspects, what could be a reasonable explanation for this difference? 

Response: The chemical profile analysis is just a qualitative analysis of the compounds present in the extracts. Therefore, the difference in the performed tests can be suggested due to the quantitative difference (concentration) of the components that would be considered as bioactive for each test performed.

- An interesting aspect, in order to think about a final application of the ethanolic extracts could be represented by the efficiency of extraction, in terms of the ratio between the initial biomass and the extracted material. Could you provide this information? Could it be possible to compare it with other different studies?

Response: Efficiency is related to solvent and extraction time. We worked with  aqueous ethanol 70% due articles reported photoprotective activity in other types of propolis such as green propolis.

When considering the cell viability, the study was performed only on the extracts and not on the final formulations too. Is there a reason for this choice to not going further?

Response: We do not perform the cell viability on the formulations, as the bases used are established standards such as Polawax without propolis. We did not know the behavior of propolis in the cell viability assay. But it is interesting in the future to test the final cell viability formulations too.

Reviewer 2 Report

The broader content of this manuscript is unquestionably of high importance and of high interest to the  to readers of IJMS. The manuscript presented has been carefully prepared, and in my opinion deserves publication in IJMS after minor revision.

Minor comments are below:

page 2 Line 52_55 Please expand on the concept of this sentence by including the importance that propolis could have in protecting plants against abiotic stress. For example, see the recently published manuscript by El-Hady et al. (DOI: 10.3390/plants10010074).

Author Response

Response for Reviewer 2

page 2 Line 52_55 Please expand on the concept of this sentence by including the importance that propolis could have in protecting plants against abiotic stress. For example, see the recently published manuscript by El-Hady et al. (DOI: 10.3390/plants10010074).

Response: Thank you and the reference has been added.

Propolis has been used as an herbal medicine and several useful activities have been described for propolis, including antiviral, antibacterial, antifungal, anti-inflammatory, antioxidant and of protection to plants against abiotic stress [1,2,3]. However, the location of the plant, bee species, seasonality, climatic differences and solvent extraction processes can affect chemical composition and biological properties [4,5,6].

 El-Hady, N.A.A.A., ElSayed, A.I., El-saadany, S.S., Deligios, P.A. and Ledda,L. Exogenous Application of Foliar Salicylic Acid and Propolis Enhances Antioxidant Defenses and Growth Parameters in Tomato Plants. Plants 2021, 10(74), 1-13.

Reviewer 3 Report

Please provide the name of country for reagents purchased from different companies. Also, in the methodology section, please provide the reference when for example stated about the Mansur method.

In my opinion,  characterization of the crude sample of red propolis should be given (e.g., whether it is a dried material, harvested when, from which areas, etc.). In line with this, please insert also both drying and storage conditions of the solution.

As the Authors used Polawax cream as a base for the propolis extract, in my opinion, some more information on its properties and biological activities should be indicated.

I was wondering whether the Authors performed and thus have some results for  skin corrosion tests utilizing reconstructed human epidermis models (RhE), or skin irritation,or sensitization. These results would improve greatly the manuscript in the aspect of propolis toxicity

When providing results for Analysis of chemical component, it would be nice if additionally the Authors insert chromatograms

Author Response

Response for Reviewer 3

Please provide the name of country for reagents purchased from different companies. Also, in the methodology section, please provide the reference when for example stated about the Mansur method.

Response: Modified.

Solvents and reagents were purchased from Synth (Diadema, SP), Vetec (Duque de Caxias, RJ) and Neon (Suzano, SP), and used without further purification. The in vitro solar protection factor (SPF) was determined by the spectrophotometric method developed by Mansur [41,42]. Absorbance readings were performed on a Genesys 105 UV-VIS spectrophotometer equipped with 1 cm quartz cell and concentration range between 0.02 and 0.1 mg/mL. The Cream sunscreen Polawax (Deionized Water-phase B, Germall 115-phase, Caprylic acid Caprylic Triglicer phase A, EDTA-phase B, Nipagim-phase B; Nipazol-Phase A, Polawax-phase A, Propyleneglicol-Phase B and BHT-phase A) was obtained by the NatureDerme Manipulation Pharmacy in Belo Horizonte, MG.

As the Authors used Polawax cream as a base for the propolis extract, in my opinion, some more information on its properties and biological activities should be indicated.

Response: Only Polawax cream is used in cosmetic formulations and therefore the research for formulations with a photoprotective cream.

I was wondering whether the Authors performed and thus have some results for  skin corrosion tests utilizing reconstructed human epidermis models (RhE), or skin irritation,or sensitization. These results would improve greatly the manuscript in the aspect of propolis toxicity

Response: This was an excellent observation. For this work, we did not carry out corrosion, irritation or sensitization tests, because, previously, our research group published an article that shows that the hydroethanolic extract of red propolis was classified as non-irritant/slightly irritating by means of the HET-Cam test. We thought that the results found from this test, for example, would be very similar to those we have already published, which would not be innovative.

Photoprotective activity and increase of SPF in sunscreen formulation using lyophilized red propolis extracts from Alagoas.

Wanessa A. da S. Almeida, Amanda dos Santos Antunes, Ricardo G. Penido, Helen S.da G. Correa, Andrea M. do Nascimento, Ângela L. Andrade, Vagner R. Santos, ThiagoCazati, Tatiane Roquete Amparo, Gustavo Henrique Bianco de Souza, Kátia MichelleFreitas, Orlando David Henrique dos Santos, Lucas Resende Dutra Sousa, Viviane M.R. dos Santos.

Revista Brasileira de Farmacognosia,

Volume 29, Issue 3,

2019,

Pages 373-380,

ISSN 0102-695X,

https://doi.org/10.1016/j.bjp.2019.02.003.

When providing results for Analysis of chemical component, it would be nice if additionally the Authors insert chromatograms

Response: Chromatograms from the chemical analysis of both extracts have been added to the supplementary material.